# Non-Wetting and Non-Reactive Behavior of Liquid Pure Magnesium on Pure Tungsten Substrates

**DOI:** 10.3390/ma15249024

**Published:** 2022-12-17

**Authors:** Sylwia Terlicka, Paweł Darłak, Natalia Sobczak, Jerzy J. Sobczak

**Affiliations:** 1Institute of Metallurgy and Materials Science, Polish Academy of Sciences, 25 Reymonta Street, 30-059 Kraków, Poland; 2Faculty of Foundry Engineering, AGH University of Science and Technology, 23 Reymonta Street, 30-059 Krakow, Poland

**Keywords:** magnesium, tungsten substrate, wettability, contact angle, sessile drop method, capillary purification procedure

## Abstract

The wetting behavior of liquid magnesium drop on pure tungsten substrates was investigated, for the first time, with the sessile drop method combined with non-contact heating and capillary purification of a Mg drop from a native oxide film. A specially designed apparatus dedicated to the investigation of the high-temperature interaction of dissimilar materials was used. The comparative experiments were performed under isothermal conditions at temperatures of 700 °C and 740 °C using two atmospheres: Ar + 5 wt.% H_2_ and pure Ar, respectively. During high-temperature tests for 180 s, the images of the Mg/W couples were recorded with CCD cameras (57 fps) from two directions of observation. The solidified drop/substrate couples were subjected to structural characterization using scanning electron microscopy (SEM) coupled with energy-dispersive X-ray spectroscopy (EDS). Under the applied measurement conditions, liquid Mg revealed non-wetting behavior on W substrates (a contact angle θ > 90°). The average value of the contact angle under the flowing Ar atmosphere at 740 °C was θ_av_ = 115°, whereas it was higher under the flowing Ar + 5 wt.%. H_2_ atmosphere at a lower temperature of 700 °C, showing θ_av_ = 122°. Independently on employed atmosphere and temperature, SEM + EDS analysis of solidified sessile drop couples did not display any new phases and mass transfer between the Mg drop and the W substrate, whereas the presence of discontinuities at the Mg/W interface of cross-sectioned couples were well-distinguished. Non-wetting and a lack of permanent bonding between the Mg drop and W substrates have a good agreement with the Mg–W phase diagram calculated with the help of FactSage software and FTlite database, i.e., the non-reactive nature of the Mg/W couple because W does not dissolve in liquid Mg and it does not form any compounds with Mg. These findings allow for the recommendation of tungsten as a suitable refractory material for long-time contact with liquid Mg in different container-assisted methods of materials characterization as well as in liquid-assisted processing of Mg components.

## 1. Introduction

Magnesium is one of the lightest and most widespread elements available on Earth. Its density (1.74 g/cm^3^) is 66% that of aluminum and 25% that of steel [1,2,3,4]. This makes it a promising lightweight material for many industries, i.e., military, aerospace, automotive, electronics, and energy storage systems due to its high strength relative to material density, suitable elastic modulus, excellent machinability, better ductility and castability than aluminum and steel, and relatively low production cost [1,2,3,4,5,6,7,8,9]. Moreover, compared to other metals and polymeric materials, magnesium is non-toxic and biocompatible [8,10]. It also has good thermal and electrical conductivity, vibration and shock absorption, and damping capacity [8,11]. These unique characteristics make magnesium a promising candidate for replacing steel and aluminum alloys in more energy-efficient and environmentally friendly applications. However, despite its good properties, pure magnesium is not widely used in the engineering or construction industries [8,12]. This is mainly due to its high affinity for oxygen, corrosion sensitivity, poor mechanical properties, and limited formability, especially at room temperature [3,8,13]. Therefore, magnesium is usually alloyed in the industry to improve its properties and increase its applicability [3,14].

However, despite the considerable efforts put into the development of Mg alloys for various applications, there is still a problem in producing homogeneous Mg alloys without oxide impurities due to some methodological problems, mainly resulting from the sublimation and evaporation of Mg at elevated temperatures and its strong affinity for oxygen. It is also a problem to find a suitable material for crucibles and molds for melting and casting magnesium and Mg-based alloys as well as for reliable measurements of their thermophysical properties by container-assisted methods, such as differential scanning calorimetry or differential thermal analysis [15], due to its high reactivity with many metals and nonmetals. Therefore, there is a great need to understand the high-temperature behavior of liquid magnesium and its alloys in contact with various refractory materials. In an inappropriate container, a molten metal can be contaminated during manufacturing, which in turn may cause structural defects in the final products (cracks, inhomogeneity, segregation, porosity, etc.).

Moreover, measurements of the thermophysical properties of contaminated materials have a large scattering and become even unreliable which may cause misleading conclusions. In addition, nowadays advanced materials engineering is based on computer simulations and predictions, which are impossible when reliable thermophysical data are not available. Hence, full knowledge of the thermophysical properties of magnesium and its alloys and their high-temperature interactions with various solid materials is of great practical importance. This knowledge is crucial especially in the liquid-assisted processing of Mg-based components, in computer-aided liquid metal engineering, and in process optimization [16,17].

For high-temperature applications, a container material should be non-wettable and non-reactive in contact with a molten metal. It is widely accepted that in high-temperature liquid/solid systems, a wetting phenomenon (a contact angle θ < 90°) may take place through two main reactive mechanisms, i.e., (1) the dissolution of the solid in the molten metal or (2) wetting through the formation of wettable reaction product at the metal/substrate interface [18]. The dissolutive wetting is easily distinguished since it is accompanied by alloying of the drop with substrate constituents and the formation of a crater in the substrate under the drop [19].

These two high-temperature phenomena play an important role in the relationships between wetting, interface structure, and bonding [19,20,21,22] and they should be considered in the analysis of high-temperature interactions between dissimilar materials because an initially *pure Me/substrate* couple is converted to a new couple *(Me-A)/Me_x_A_y_* (*A*—substrate; *(Me-A)*—a solution of A in liquid Me; *Me_x_A_y_*—interfacial reaction product), whereas a deep crater may cause an apparent value of contact angle measured by the commonly used sessile drop method.

Following available literature data [23,24,25] on the phase diagrams of Mg with transition metals of groups 5–8 (i.e., Mo, Ta, or W), it can be concluded that these metals may be suitable as refractory materials for long-time contact with molten Mg because they neither form intermetallic phases with Mg nor dissolve in liquid Mg, and very often the molten alloys of Mg with those elements have the miscibility gap. For these reasons, in previous studies, Terlicka et al. [26] and Dębski et al. [27] applied W as container materials in the measurements of thermodynamic properties of liquid Mg-based alloys.

Taking into consideration a lack of experimental data and information in the literature on high-temperature interactions of Mg and its alloys with such metals, while the existing phase diagrams for them are mostly only calculated and require verification, the present work focuses on an experimental investigation of the factors affecting wetting, interface structure, and bonding between pure tungsten substrates and liquid pure Mg. This type of study was carried out for the first time, and the results obtained will undoubtedly expand the negligible knowledge of high-temperature interactions of liquid Mg with dissimilar materials as reported in [16,28,29,30,31,32,33,34,35,36,37,38,39]. In addition, most of the available data is dedicated to high-temperature measurements of pure magnesium with ceramics [16,28,30,32,34,35,36,37,38,39], whereas only three of them undertake the topic of the interaction of Mg with metallic substrates (Ni, Ti, steel, and Fe) [29,31,33].

The results presented in this paper also allow making a conclusion on whether tungsten can be used as a material for containers, crucibles, or molds for melting and casting magnesium and magnesium alloys. This conclusion is supported by the analysis of the Mg–W phase diagram recalculated in this study because the only information that exists is from the 20th century. As reported by Nagender-Naidu and Rama Roa [40] and Predel [41], Kremer [42], as well as Sauerwald [43] could not detect any reaction between W and Mg. Busk [44] found that the lattice parameters of Mg are not modified by the addition of W. Therefore, the present study indirectly confirms that the recalculated phase diagram is reliable at the temperatures used in this work.

## 2. Materials and Methods

### 2.1. Apparatus for High-Temperature Experiments

The wettability of a metallic tungsten substrate (Wolften, 99.99 wt.%) with liquid magnesium (>99.9 wt.%) at high temperature was tested with the sessile drop method using a new specially designed apparatus [45] shown schematically in Figure 1.

This device allows real-time observations and measurements in a variety of gaseous environments or high vacuum at temperatures up to 2100 °C. The apparatus, apart from investigating the high-temperature interactions of different materials (metallic or non-metallic) by means of the conventional sessile drop method combined with a contact heating procedure, enables in a single test to apply several different methods and test procedures (such as non-contact heating of a couple of dissimilar materials, drop cleaning by capillary purification, etc.).

The specially designed heating system makes it possible to use two independent heaters made of different materials and to produce different atmospheres inside the test chamber. Moreover, the heaters can operate independently of each other, allowing measurements under isothermal, as well as non-isothermal conditions.

In order to get maximum information about an object subjected to high-temperature observations, the apparatus is equipped with two high-resolution, monochromatic, high-speed CCD cameras with a Tamron add-on lens that allows real-time simultaneous imaging of the experiment from two perpendicular directions. The first camera is positioned perpendicular to the cross-section of the apparatus shown in Figure 1a, and a second camera is placed parallel to that position (on the left side of Figure 1a). Both cameras are equipped with filters that prevent a negative effect of the light generated from the studied object. The first camera has external lighting, which is additionally enhanced by a blue LED to improve the contrast of the recorded images (Figure 1b). Two CCD cameras placed perpendicular to each other allow positioning of the liquid drop in the center of the substrate, checking drop symmetry, and more detailed observation of high-temperature phenomena taking place with the drop/substrate interaction.

The apparatus is equipped with a set of four K-type thermocouples (two dedicated to the capillary heater and two for the substrate heater). In the first set of two thermocouples (dedicated to the capillary heater), one is located directly at the capillary (TC1, Figure 1c) and the other is located right next to the capillary heater. In the second set, one of the thermocouples is located in the direct area of the heater, and the other is placed in the manipulator on which the substrate samples are provided (the thermocouple TC2 is located directly under the test substrate, Figure 1c). Temperature detection from the thermocouples located directly at the heaters (both the one from the capillary and the one from the substrate) is carried out using Eurotherms placed in the furnace power supplies. Whereas the temperature values from thermocouples placed directly at the capillary with liquid Mg and at the substrate are collected using the PICO Technology system (TC-08 thermocouple data logger), which allows real-time data collection and display. Moreover, the positioning of the thermocouples allows the control of heating/cooling rates and monitoring of the temperature field around the drop/substrate couple.

### 2.2. Wettability Tests

Before the measurements, the surface of the Mg rod with a diameter and length of about 10 mm was mechanically ground with SiC papers, followed by ultrasonic cleaning in isopropanol (STANLAB, p.a.) for approximately 10–15 min. The Mg rod thus cleaned and degreased was then placed in a graphite capillary and immediately transferred to the experimental chamber. The surface of the tungsten substrates (a rod-shape with a diameter of 10 mm and a thickness of 5 mm) was polished with sandpapers with a gradation of 200 to 4000, followed by final polishing with diamond suspensions from 3 to 1/4 µm and anhydrous SiO_2_ suspensions to obtain a mirror-finished surface (Figure 2a). Directly before testing, the W substrate was also ultrasonically cleaned in isopropanol for 20 min. Using isopropanol is a common practice for sample preparation for high-temperature wettability tests because it evaporates during pumping and does not affect wetting behavior.

A high-temperature experiment of wetting a solid tungsten substrate with liquid magnesium at a temperature of 740 °C was conducted in the pure Ar atmosphere, whereas at 700 °C, the reductive Ar + 5 wt.% H_2_ atmosphere was used to check the possible effect of the applied atmosphere on wetting behavior and contact angle values of the Mg/W couple during holding at the test temperature.

After the samples were placed in the chamber, a vacuum of 10^−6^ mbar was created, and then an inert gas (Ar 99.999 wt.% or Ar + 5 wt.% H_2_) was passed through the chamber (flowing gas, a pressure between 1.0 × 10^3^–1.1 × 10^3^ mbar). For the experiment performed in the Ar + 5 wt.% H_2_ atmosphere, after the gas was let into the chamber, the chamber was again evacuated to a vacuum of 10^−6^ and again the gas was let in to flush the chamber. The process was repeated once again. Additionally, the test carried out in Ar + 5 wt.% H_2_ was performed in the presence of Ti chips and a sponge, as well as a piece of pure Zr (99.98 wt.%), which were placed next to the W substrate and functioned as oxygen getters [15].

To eliminate the effect of a native oxide layer on the Mg drop on contact angle measurements, the non-contact heating of a tested couple of materials to the test temperature was combined with in situ mechanical cleaning of the drop from a native oxide film directly in a high-temperature chamber using a capillary purification (CP) procedure [15,28]. For this, after reaching the desired testing temperature, the Mg drop was mechanically squeezed through a hole in the graphite capillary and deposited on the substrate, as it is schematically presented in Figure 3. It should be noted that previous studies [16,28] have evidenced that graphite is non-wettable and non-reactive in contact with liquid Mg. In addition, the use of the CP procedure makes the contact angle measurements more reliable, compared to the traditional contact heating procedure, because the drop produced in the CP procedure has a perfectly symmetrical shape.

After squeezing and placing the drop on the substrate, the Mg/W couple was left at the set temperature for about 180 s. During the experiments, images of the Mg/W couple were captured with two CCD cameras at the same speed of 57 frames per second. Only images recorded with the first monochromatic CCD camera with additional LED lighting were then processed to measure contact angle values and analyze wetting kinetics with time using a drop shape analysis program which fits a contact angle using the polynomial fitting [46,47]. This program was modified for grayscale images and implemented into MATLAB software (R2022b version).

### 2.3. Microstructure Observations

The microstructure observations coupled with the analysis of the local chemical composition of the sessile drop Mg/W couples were performed using an FEI E-SEM XL30 Scanning Electron Microscope (SEM) integrated with an energy dispersive X-ray spectrometer EDAX GEMINI 4000 (EDS). For this, each Mg/W couple was immersed in thermosetting epoxy resin, cut in half in the central part of a solidified drop, and then ground on sandpapers with a gradation from 200 to 4000. The samples prepared in this way were polished using anhydrous 3 µm to 1/4 µm diamond and anhydrous SiO_2_ suspensions. On top of all obtained metallographic samples, a thin carbon layer was deposited (about 20–50 nm) to secure good electric conductivity during the SEM observation, as well as to avoid oxidation of the samples. Despite using soft, anhydrous polishing suspensions, it was not possible to obtain perfectly smooth surfaces for observation due to the large difference in hardness between Mg and W. Microstructure observations and chemical composition analyses were performed at an accelerating voltage of 20 kV with a spot size of 4.5 µm, a working distance of 10 µm, and magnifications from 50× to 10,000×.

## 3. Results and Discussion

Selected images of the liquid Mg/W couples, corresponding to important stages of sessile drop tests combined with the capillary purification procedure (drop squeezing, drop deposition, detachment of the graphite capillary from the drop, the end of the test), recorded during holding at 740 °C and 700 °C, are presented in Figure 4a and Figure 4b, respectively.

As can be seen by analyzing the recorded images (Figure 4a,b), the squeezed Mg drops had a regular and spherical shape, and the perfectly symmetric shape of the Mg drop deposited on the W substrate was maintained until the end of the experiment. The drop surface for both experiments was shiny during isothermal heating as can be seen on the drop images recorded with a second CCD camera (Figure 5). On the contrary, in the contact heating procedure, the Mg drop surface is usually matte because of a thick primary native oxide film covering the whole drop surface [28]. For many metals sensitive to oxidation, the presence of such a primary oxide film on a sample prevents the formation of the drop or influences its symmetry and may avoid the direct contact between the substrate and the drop, and consequently, affects wetting behavior and the value of the contact angle [15,28,29].

Intense evaporation (especially well visible when the external light is turned off at the start of cooling) of liquid magnesium after the dropping (see Figure 5b) can cause a reaction of the Mg with the residual oxygen present in the measurement chamber, and the secondary surface oxidation of the Mg drop, hence the lack of a perfectly shiny surface of the drop as seen in Figure 2b. In the case of a drop squeezed in a reductive atmosphere, where the addition of hydrogen decreases the oxygen partial pressure in the chamber [15], the obtained drop is shinier and the secondary oxidation during cooling is less pronounced.

From the images obtained with the first CCD camera (e.g., in Figure 4), the left, right, and average values of the contact angle (θ_l_, θ_r_, θ_av_, respectively) were determined with the use of the polynomial fitting using the drop shape analysis program script [47]. The program determines the contact angle based on the obtained images of Mg/W couples by fitting a polynomial of the fourth degree [46]. A step-by-step implementation of the contact angle fitting algorithm is shown in Figure 6 (taken from MATLAB software) [47].

Figure 7 shows the kinetics of wetting as the change in the contact angle values θ_l_, θ_r,_ and θ_av_ versus time. Under the flowing Ar atmosphere, θ_av_ = 115°, i.e., lower than that for the test performed under the flowing Ar + 5 wt.% H_2_ atmosphere, where θ_av_ = 122°. The slight fluctuations in contact angle values seen in the points in Figure 6 are caused by the oscillation of the Mg drop after deposition on the W substrate and because of a slight drop movement due to the evaporation of Mg (Figure 7a,b).

Independently on the type of applied atmosphere, all values of the contact angles are θ > 90° and they did not change with time during the course of the sessile drop tests, thus showing the non-wetting behavior of liquid Mg on W substrates. It is not possible to state clearly whether there is a significant effect of the atmosphere on contact angle values here. Usually in liquid metal/solid systems, the higher the applied temperature, the lower the contact angle value.

Following the reports of Shi et al. [33], focused on the experimental study of Mg/Fe, Mg/SUS430-steel, and Mg/alumina couples, we may conclude that temperature might be a very strong factor that significantly reduces the contact angle values of liquid Mg on different solids because only a 25 °C increase in temperature (from 700 °C to 725 °C) caused in 120 s contact with the same substrates, respectively, 25°, 60°, and 40° decrease in contact angle values. Shi et al. [33] explained this strong effect by the intensive evaporation of Mg, which results in a reduction in the volume of the drop and thus an apparent decrease in the contact angle value with time.

A similar explanation for the unusual change in the contact angle in the Mg/MgO system was given by Fujii et al. [35]. Therefore, we suggest that in our study, the 7° difference in contact angle values measured in the Ar + H_2_ atmosphere vs that in pure Ar is rather related to the 40 °C difference in temperatures in these tests.

Figure 8 and Figure 9 show the representative results of the structural characterization of solidified and cross-sectioned Mg/W couples. SEM + EDS observations did not show the presence of any new phases at the interfaces, as well as mass transfer through the interface and mixing of the components. Moreover, the discontinuities observed at the contact between the Mg drop and the W substrate (e.g., well-distinguished in the right edge and propagating along the Mg/W interface in Figure 8c and Figure 9c) evidence a lack of good bonding due to no interaction of liquid Mg with the W substrate.

As mentioned above, the preparation of high-quality metallographic specimens of Mg/W couples is exceptionally problematic because of the extremely different hardness of Mg and W coupled with the possible fast oxidation of Mg that may cause overinterpretation of structural observations. During polishing, the magnesium surface polishes (grinds) much faster than the W, resulting in a “step” and cracks at the Mg drop near the Mg/W interface and/or at the top of the drop. Contaminants from the polishing paste very often collect in this area and it is difficult to remove them from this place. The effect of the sample preparation history is particularly noticeable in Figure 8b–e and Figure 9c–e, where many cracks and losses are visible on the solidified drop.

The comparison of the results of this study on wetting and bonding behaviors of Mg/W couples with those of structural characterization of interfaces formed between liquid Mg and W substrates allows us to make a conclusion on the specific high-temperature interaction of these two dissimilar materials.

Despite the fact of a lack of detailed experimental phase diagram of the Mg–W system in the Mg-rich region, the one calculated in this study, with the help of FactSage software [23,24,25] and presented in Figure 10a,b, is consistent with our above-mentioned experimental findings. Following the calculated Mg–W phase diagram, Mg does not form with W any compounds in the entire concentration range, whereas in the liquid state corresponding to the experimental temperature range used in this study (T_exp_ = 700–740 °C), liquid Mg shows a complete lack of W dissolution (Figure 10b).

## 4. Conclusions

For the first time, the high-temperature interaction of liquid magnesium with solid tungsten has been studied experimentally with the sessile drop method accompanied by detailed structural observations of solidified Mg/W couples and analysis of the Mg–W phase diagram calculated with the help of FactSage software and FTlite database.

The sessile drop tests performed at two temperatures of 740 °C and 700 °C in two protective atmospheres (pure Ar and Ar + 5 wt.% H_2_, respectively) using a newly designed test apparatus, allowing to apply improved testing procedure accompanied by the simultaneous observation of the drop/substrate couples in two directions using dissimilar recording modes, showed that under the applied testing conditions, liquid Mg free of a native oxide film exhibits non-wetting (θ > 90°) and week bonding with the W substrate.

Structural characterization of solidified sessile drop couples with SEM + EDS analysis did not evidence any new phases and mass transfer through the Mg/W interface as well as numerous discontinuities between the Mg drop and W substrates.

Non-wetting and weak bonding in the Mg/W couples observed under the conditions of this study are in good agreement with the calculated Mg–W phase diagram showing the non-reactive nature of the Mg–W system because W does not dissolve in liquid Mg and it does not form any compounds with Mg.

The above findings allow recommending of tungsten as a suitable refractory material for containers, crucibles, casting molds, and other metallurgical and experimental appliances in different container-assisted processing and testing methods. The non-wetting and non-reactive character of molten Mg with respect to W, especially the near-zero solubility of W in molten Mg at industrially important temperature ranges (<750 °C), makes W particularly appropriate for long-time contact with liquid Mg (e.g., taking place in directional solidification of Mg melt in GASAR process [48]) or in long-time storage of liquid Mg.

## Figures and Tables

**Figure 1 materials-15-09024-f001:**
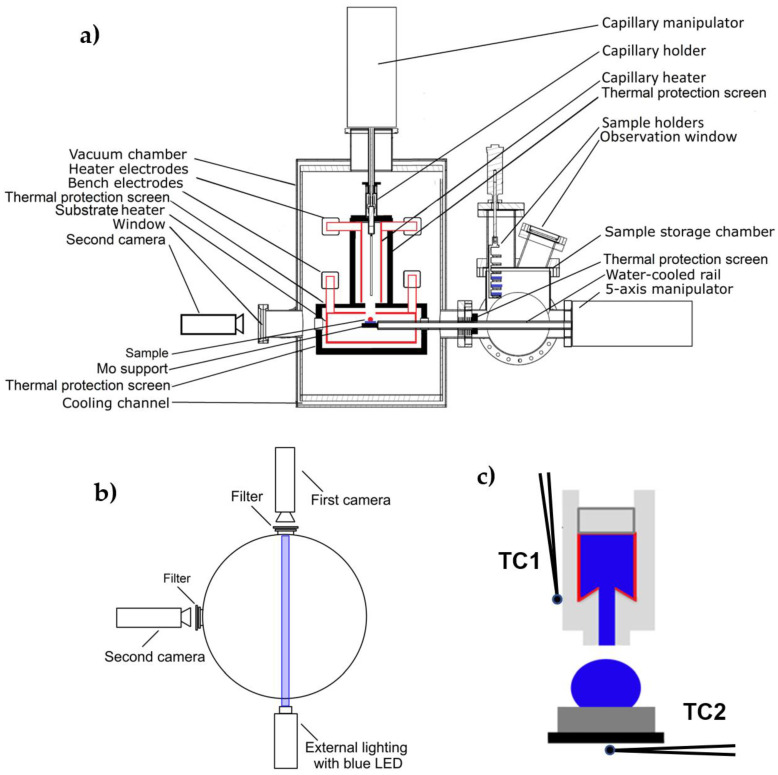
Schematic presentation of the apparatus used for real-time observation of high-temperature phenomena taking place during the interaction of molten materials with dissimilar substrates [45]: (**a**) side-view, (**b**) top-view, (**c**) location of thermocouples at the capillary (TC1) and under the test substrate (TC2).

**Figure 2 materials-15-09024-f002:**
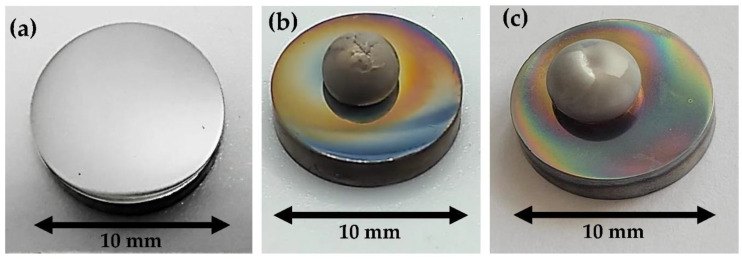
Side-view photos of tungsten substrates before (**a**) and after (**b**,**c**) the wettability experiments in the Ar (**b**) and in the Ar + 5 wt.% H_2_ (**c**) atmosphere.

**Figure 3 materials-15-09024-f003:**
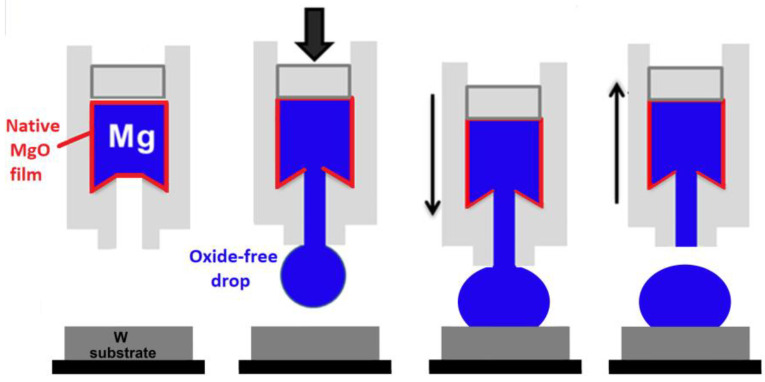
A scheme of the high-temperature wettability test using the sessile drop method with non-contact heating and capillary purification procedure.

**Figure 4 materials-15-09024-f004:**
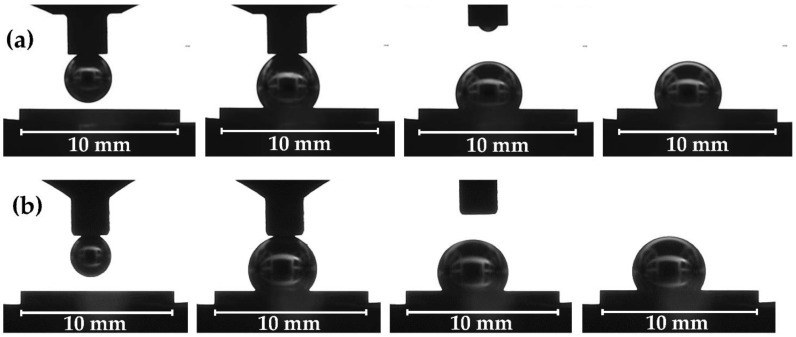
Images recorded with the first monochromatic CCD camera with external LED lightening during the CP sessile drop tests of Mg/W couple: (**a**) at T = 740 °C, under flowing Ar atmosphere, (**b**) at T = 700 °C under flowing Ar + 5 wt.% H_2_ atmosphere; the last image for both sets corresponds to the end of the test after isothermal contact for 180 s.

**Figure 5 materials-15-09024-f005:**
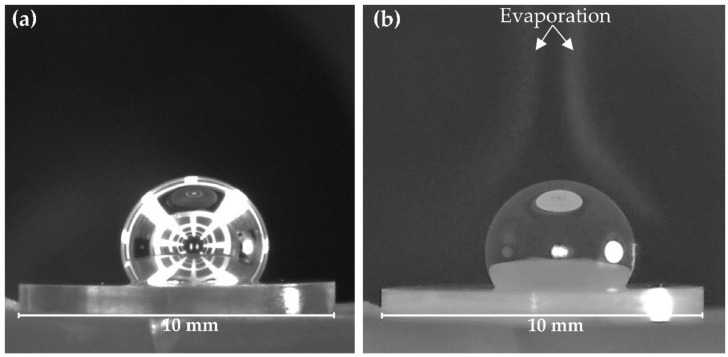
Images recorded with the second CCD camera during the wettability test of Mg/W couple under flowing Ar + 5 wt.% H_2_ atmosphere at T = 700 °C: (**a**) after deposition; (**b**) after 180 s, once cooling has begun.

**Figure 6 materials-15-09024-f006:**
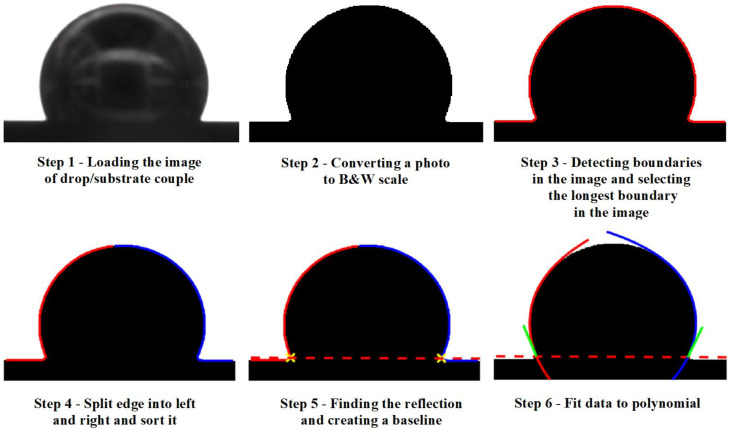
Step-by-step implementation of the contact angle fitting algorithm using the drop shape analysis program script [46,47].

**Figure 7 materials-15-09024-f007:**
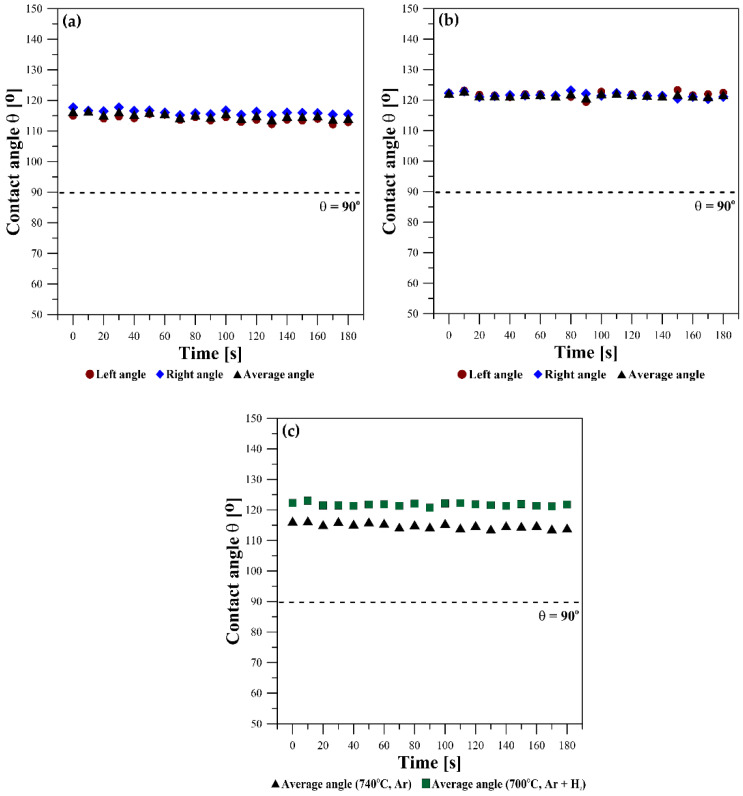
Wetting kinetics of liquid Mg on the tungsten substrate registered for 180 s under different atmospheres: (**a**) Ar at T = 740 °C; (**b**) Ar + 5 wt.% H_2_ at T = 700 °C; (**c**) comparison of θ_av_ between Mg/W couples received under the flowing Ar at T = 740 °C and Ar + 5 wt.% H_2_ at T = 700 °C, respectively.

**Figure 8 materials-15-09024-f008:**
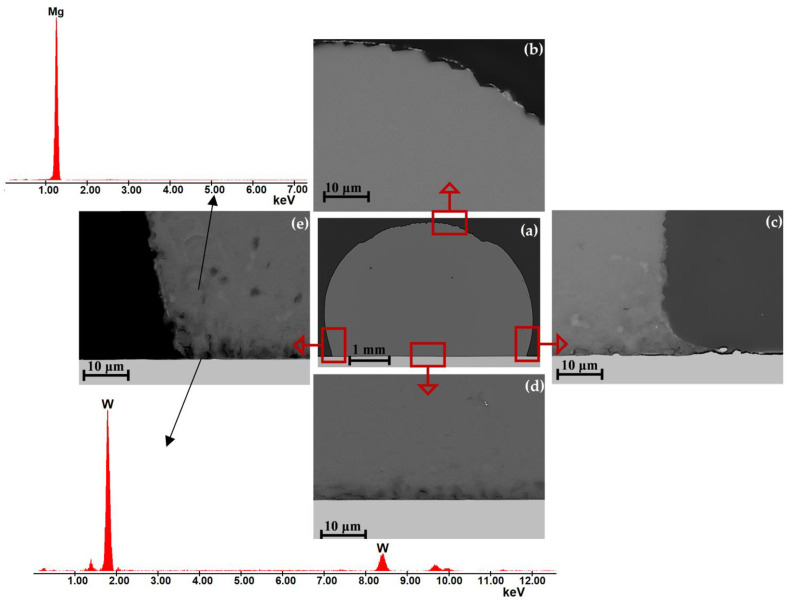
SEM images of a cross-sectioned Mg/W couple (CP: 740 °C, 180 s, Ar) taken with a BSE detector at (**a**) magnification 50× and (**b**–**e**) magnification 5000×; (**b**) top of the drop, (**c**) right drop edge, (**d**) drop/substrate interface, and (**e**) left drop edge with EDS spectra of spot analysis.

**Figure 9 materials-15-09024-f009:**
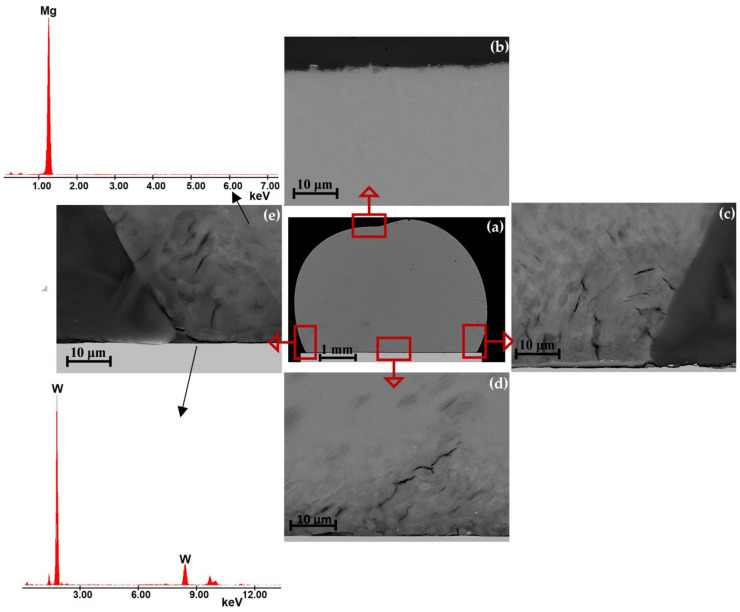
SEM images of a cross-sectioned Mg/W couple (CP: 700 °C, 180 s, Ar + 5 wt.% H_2_) taken with a BSE detector at (**a**) magnification 50× and (**b**–**e**) magnification 5000×; (**b**) top of the drop, (**c**) right drop edge, (**d**) drop/substrate interface, and (**e**) left drop edge with EDS spectra of spot analysis.

**Figure 10 materials-15-09024-f010:**
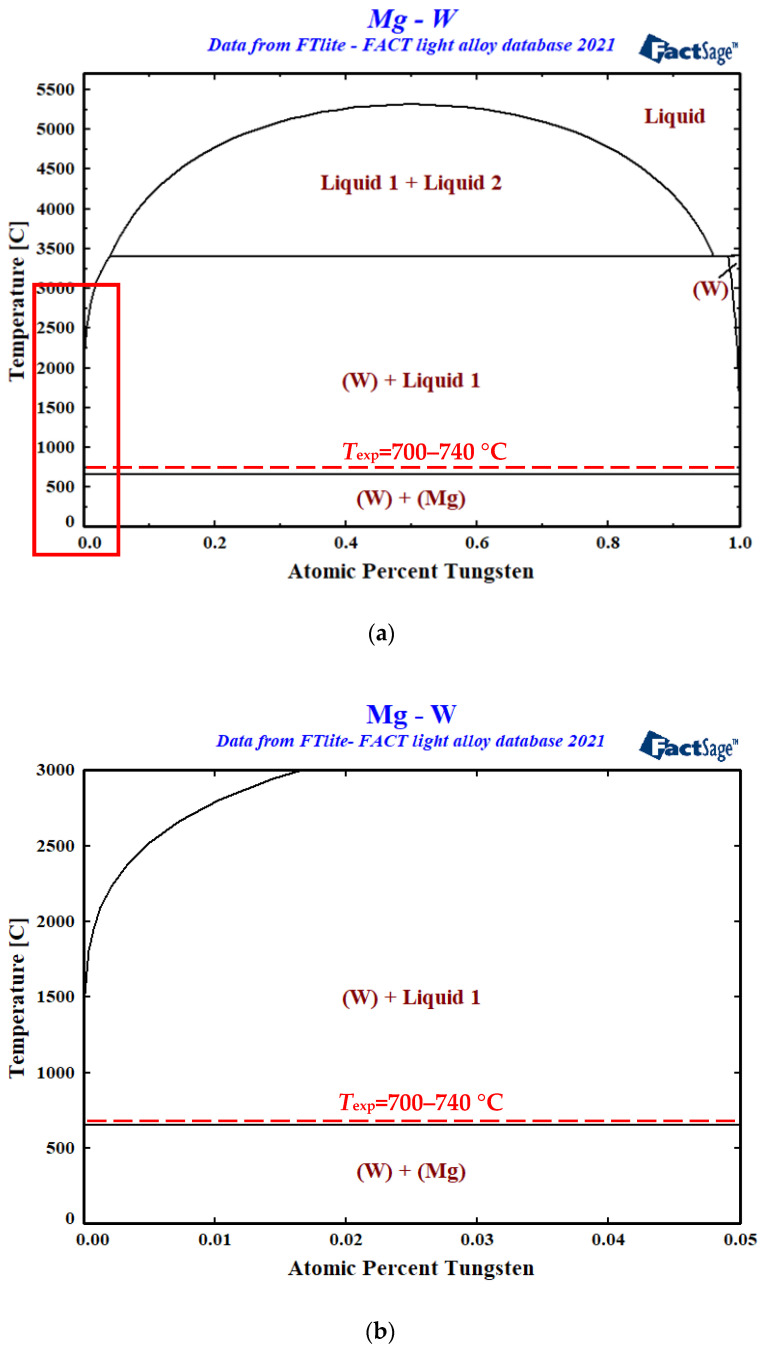
Phase diagram of the Mg–W system calculated in this study with the use of FactSage software and FTlite database [23]; the insert in (**a**) corresponds to the magnified Mg-rich region shown in (**b**).

## Data Availability

Raw data is available upon request.

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
