# Peer review of "Non-Wetting and Non-Reactive Behavior of Liquid Pure Magnesium on Pure Tungsten Substrates"

_materials, 2022, doi:10.3390/ma15249024_

Round 1
Reviewer 1 Report
In this paper, the authors investigated the wetting behavior of liquid magnesium drop on pure metal tungsten substrates, the specially designed apparatus dedicated to investigation of the high-temperature interaction of dissimilar materials was new. So, I suggest that this paper can be published in this journal.
Reviewer 2 Report
The study employed sessile drop method to probe the high-temperature interaction of liquid magnesium with solid tungsten, and revealed that a non-wetting and weak bonding between liquid magnesium and tungsten substrate. I have a few questions and comments:
Is that possible to show the contact angle fitting profile generated by the Drop Shape Analysis Program in MATLAB?
(page 3) "at a temperature of 740 °C was conducted in the pure Ar atmosphere, while 108 at 700 °C, the reductive Ar + 5 wt.%" , how the two temperatures were determined? Are they from any references?
The author used non-contact heating method to avoid the formation primary oxide film on the drop. Can the author use characterizations to prove the native oxide film is effectively avoided?
(page 3) the surface of the Mg and W substrate were ultrasonically cleaned in isopropanol. Will isopropanol affect the wetting behavior? Or it will be evaporate at high temperature?
Can the author add scale bars on Figure 4 and 5?
Reviewer 3 Report
This article is attractive for readers and has scientific innovation. The presented materials and their results are scientifically and accurately expressed, and in general, I suggest the publication of this article. However, some minor modifications are needed:
- Authors should add the most important results of this research quantitatively in the abstract.
- For a better understanding of the readers, the authors should add explanations in the introduction section for each of the references.
- In addition, the order of presenting the references should be such that the reader's mind is directed towards the innovation of the article.
- The main innovation of the article compared to similar research should be added at the end of the introduction section.
- In my opinion, schematic figure 3 is redundant and does not provide any useful scientific information. Authors can remove it.
- The discussion on the results should also be strengthened.
Reviewer 4 Report
The article is written at a high level. The methodology is described in sufficient detail, the conclusions are substantiated.
Some changes can be made to improve the perception of the study by readers.
Your overview is very short and quite general. It would be nice to expand it. Describe why such experiments are needed and what they give in practical terms, as you do in the conclusions (describing W as a material for a lining when melting Mg alloys). On lines 212–221, you give for comparison the results of studies by other authors. It would be good to include similar studies in the narrow area of ​​your specialization in the Introduction.
You also write in your Conclusions that the results of your work correspond to the calculated Mg-W phase diagram. Why then experiment, if there is already a given phase diagram? It would be good to explain this in the same section.
In the Introduction, write in detail about the lack of experimental data and refer to this calculated phase diagram.
You have a reference to the results of the phase analysis. Why not present the results of this phase analysis in the article and also present the results of chemical analysis, proving that metals do not interact?
Round 2
Reviewer 4 Report
The authors corrected article. I thank them for their work. The newly added figures could use a slightly larger font. Then it would be easier to read the inscriptions in the figure.
